# Activation of stably silenced genes by recruitment of a synthetic de-methylating module

Wing Fuk Chan [1,2,5] ✉, Hannah D. Coughlan[1,2], Yunshun Chen[1,2], Christine R. Keenan[1,2], Gordon K. Smyth[1,2,3], Andrew C. Perkins[4], Timothy M. Johanson[1,2] & Rhys S. Allan[1,2] ✉

Stably silenced genes that display a high level of CpG dinucleotide methylation are refractory to the current generation of dCas9-based activation systems. To counter this, we create an improved activation system by coupling the catalytic domain of DNA demethylating enzyme TET1 with transcriptional activators (TETact). We show that TETact demethylation-coupled activation is able to induce transcription of suppressed genes, both individually and simultaneously in cells, and has utility across a number of cell types. Furthermore, we show that TETact can effectively reactivate embryonic haemoglobin genes in non-erythroid cells. We anticipate that TETact will expand the existing CRISPR toolbox and be valuable for functional studies, genetic screens and potential therapeutics.

Clustered regularly interspaced short palindromic repeats and the associated Cas9 endonuclease (CRISPR/Cas9) represent a transformative and programmable tool to modify the genome[1]. Through Watson-Crick base pairing, the RNA-guided Cas9 can target the genome ubiquitously, as long as a very short protospacer adjacent motif (PAM) is present. Cas9 was further engineered to remove nucleolytic activity (dCas9) and repurposed as a DNA-binding platform[1–3]. As such, gene transcription can be induced by recruiting transcriptional activators to dCas9 via direct fusion or indirect tethering. While fusion of a single activation domain VP64 causes only modest gene upregulation[4,5], the second generation CRISPR activators involve recruitment of multiple effectors, of which the dCas9-VPR[6], SunTag-VP64[7] and synergistic activation mediator[8] (SAM) appear to be the most potent systems[4].

Programmable gene activation has led to a plethora of applications, including dissection of gene function[1,3,9], genetic screening for important coding or non-coding elements[1,3,9], programmed cellular differentiation[6] and curative therapeutics[1,3,9]. Such applications require the robust activation of candidate genes regardless of the repressive

elements present at the relevant loci, including DNA methylation[10]. Thus, any system that can expand our ability to remove or circumvent these repressive elements has obvious value.

Here we demonstrate the suboptimal potency of second-generation activators SAM and SunTag-VP64 in activating deeply silenced genes that are DNA methylated. To circumvent it, we devise the TETact system by coupling the DNA demethylating factor TET1 with transcriptional activators. This improved tool activates heavily suppressed genes that are otherwise refractory to the current CRISPR activators. We demonstrate the potency in activating various genes, in different cell types and the ability of multiplexed targeting.

## Results

### Development of a TET1-based system to activate silenced genes
In a previous study, we characterised a long non-coding RNA species *Dreg1* within the enhancer region of *Gata3*[11]. Expression of *Dreg1* is highly correlated with *Gata3* expression being expressed in T-cell subsets, but completely and stably silenced in B cells. To gain insight into *Dreg1* function, we attempted to activate it in a murine B cell line

[1]The Walter and Eliza Hall Institute of Medical Research, Parkville, VIC 3052, Australia. [2]Department of Medical Biology, The University of Melbourne, Parkville, VIC 3010, Australia. [3]School of Mathematics and Statistics, The University of Melbourne, Parkville, VIC, Australia. [4]Australian Centre for Blood Diseases, Monash University, 99 Commercial Rd, Melbourne, VIC 3004, Australia. [5]Present address: Australian Centre for Blood Diseases, Monash University, 99 Commercial Rd, Melbourne, VIC 3004, Australia. ✉e-mail: luke.chan@monash.edu; rallan@wehi.edu.au

(A20) using second-generation CRISPR activation systems, SAM and SunTag-VP64. Unfortunately, targeting the *Dreg1* transcription start site (TSS) with either SAM or SunTag-VP64 failed to activate transcription (Fig. 1a).

Interestingly, activation of other lncRNAs using the second-generation CRISPR activators only leads to very low or modest upregulation[4,8] and we postulated that DNA methylation may be an impediment to efficient activation of these genes[12]. The DNA methylation pattern of the *Dreg1* locus in T and B cells was determined via publicly available whole genome bisulphite sequencing (WGBS) data[13]. As predicted, regions around the *Dreg1* TSS and gene body are differentially methylated (Fig. 1b) between the two cell types, with most CpG dinucleotides in B cells being heavily methylated.

This prompted us to investigate the possibility of activating a heavily methylated and repressed *Dreg1* by simultaneously recruiting the DNA demethylating enzyme TET1[14] and transcription activators to the target site. A recent study utilised a direct fusion of the catalytic domain of TET1 (TET1CD) to dCas9 to reactivate synthetically silenced genes[15]. However, due to the large size of TET1CD, direct fusion to dCas9 together with a selection marker is unfavourable in the context of immune cells or for therapeutic application, as it likely exceeds the cargo limit of lentiviral vectors. In addition, previous studies have suggested that more efficient gene activation is achieved with multiple copies of TET1CD[16]. We therefore adopted the previously described SunTag approach for the recruitment of TET1CD[16], and the RNA aptamer MS2 harboured within the sgRNA for the recruitment of

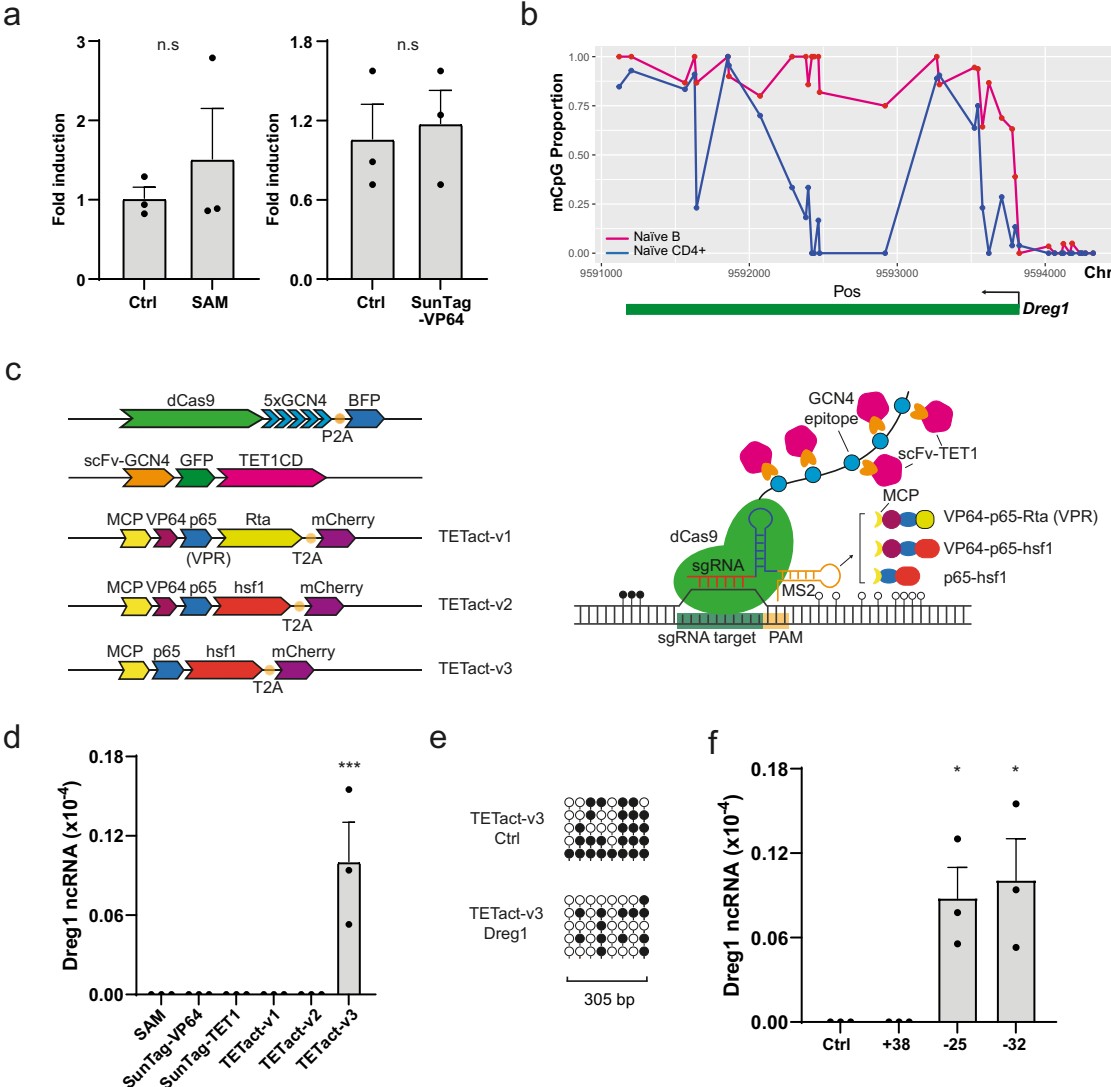

**Fig. 1 | Activation of the T-cell specific lncRNA *Dreg1* in A20 B cells. a** Fold activation of *Dreg1* in A20 cells transduced with sgRNA targeting *Dreg1* promoter together with SAM or SunTag-VP64 constructs. Fold change is calculated by ΔΔCt method. Data were analysed with unpaired two-sided Student's *t*-test compared to control **b** DNA methylation (mCpG) profiles of naïve B and CD4 + T cells at the *Dreg1* locus, plotted as population proportion of methylated cytosine in each CpG dinucleotide motif. **c** Schematics of TETact systems and corresponding construct designs−multiple copies of TET1CD are recruited to dCas9 via the GNC4 epitopes, whereas the activator domains (v1−VPR, v2−VP64-p65-hsf1, v3−p65-hsf1) are recruited via two MS2 aptamers. **d** *Dreg1* lncRNA expression in A20 cells transduced with sgRNA targeting *Dreg1* promoter in different activation systems as indicated.

*P* = 0.0003 from one-way ANOVA with Dunnett's *post hoc* test compared to SAM. **e** Bisulphite sequencing of *Dreg1* TSS and promoter for A20-TETact-v3 cells transduced with either control or *Dreg1*-targeting sgRNA. Open lollipops represent non-methylated CpG whereas closed lollipops represent methylated CpG motif. Each row represents an individual clone. **f** Activation of *Dreg1* lncRNA using different sgRNA targeting location. Expression level is relative to β-actin (*Actb*) level as 2^−ΔCt. From left to right, *P* = 0.0248, 0.0124 from one-way ANOVA with Dunnett's *post hoc* test compared to control. Data shown are mean ± s.e.m. from three independent transductions. n.s., non-significant, \**P* < 0.05, \*\*\**P* < 0.001 Source data are provided as a Source Data file.

different combinations of transcription activators herein designated as TETact (Fig. 1c, TETact v1-v3). Of the three combinations tested, the fusion of MS2 coat protein with the bipartite activator (p65-hsf1, v3) are most effective in inducing *Dreg1* transcription, from an undetectable level in A20 controls, expression was significantly upregulated to 1/100000 of β-actin level (Fig. 1d). Surprisingly, recruitment of tripartite activators (VP64-p65-hsf1 or VPR) failed to activate the lncRNA, possibly due to steric hindrance imposed by the larger size of these tripartite activators (Fig. 1d). Bisulphite sequencing of the *Dreg1* TSS and promoter region has confirmed the successful DNA demethylation of the region, which is of around 300 bp, in TETact-v3 A20 cells (Fig. 1e).

Next, we tested the effects of module position on activation strength by designing sgRNAs targeting 3 different sites around the TSS (Fig. 1f). As predicted, activation is extremely sensitive to the target site location in relation to the TSS. While sgRNAs located upstream of the TSS robustly induced *Dreg1* expression, activation did not occur when the sgRNA target site was towards downstream of the TSS (Fig. 1f). Given that the TSS of many lncRNAs and enhancer RNAs are

poorly annotated, these experiments suggest caution, as mistargeting by only 10 s of base pairs can cause failure of activation.

## Rapid, stable and specific gene activation by TETact

To further characterise TETact, we next performed a detailed assessment of the efficiency and kinetics of activation of CD4, a surface protein that defines a subset of T cells. Again utilising WGBS data[13], a differential DNA methylation pattern was observed at the *Cd4* promoter between B and T cells (Supplementary Fig. 1), with the highly methylated DNA in B cells consistent with the lack of transcription of *Cd4* in this cell-type. We designed sgRNAs targeting the *Cd4* promoter in A20 cells using different CRISPR activation systems and monitored expression by flow cytometry for up to 14 days (Fig. 2a). As predicted, second-generation activators failed to drive a high level of CD4 expression (Fig. 2b, c and Supplementary Fig. 2). As such, the SAM population showed modest levels of detectable surface CD4 expression on day 4 (MFI ~ 500), which was significantly higher than the control population expressing non-targeting sgRNA (*P* < 0.01 vs control, *t*-test). Similarly, SunTag-VP64 cells showed minimal detectable

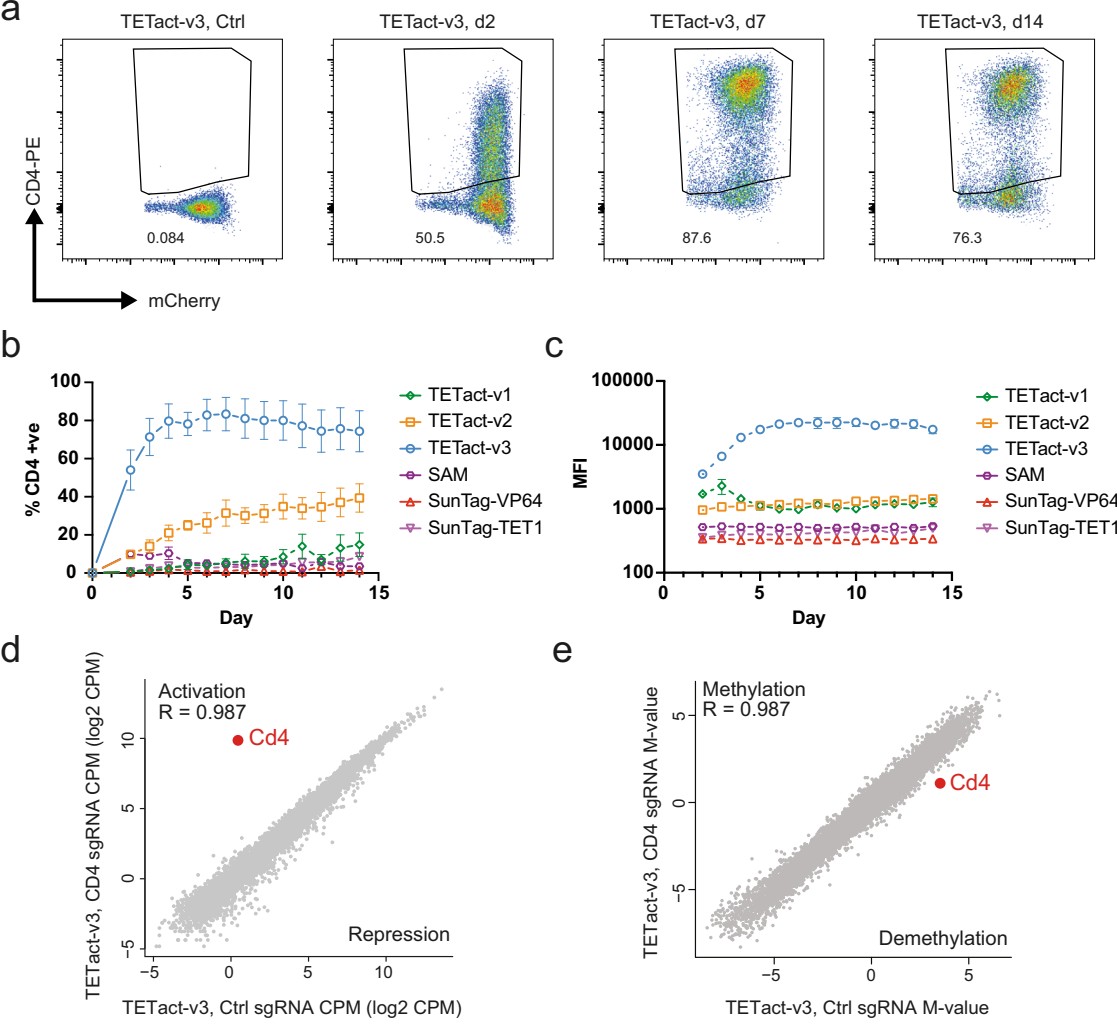

**Fig. 2 | TETact is a potent activator of stably silenced genes. a** Representative flow cytometry plots showing CD4 surface expression in A20-TETact-v3 cells transduced with *Cd4* promoter-targeting sgRNA on the indicated day post-sgRNA-transduction. Positive gates for each time point were set against the negative population of cells transduced with control sgRNA on the same day. **b** Percentage of population with surface CD4 expression over a 14-day time course for A20 cells with various activators as indicated. Data shown are mean ± s.e.m. from three independent experiments. **c** Median fluorescence intensity (MFI) of CD4-PE of the

CD4 + population over a 14-day time course. Data shown are mean ± s.e.m. from three independent experiments. **d** Gene expression (log₂CPM) or **e** DNA methylation level (*M*-value) of promoter in cells with *Cd4* promoter-targeting sgRNA versus gene expression in cells transduced with non-targeting control sgRNA. Transduced cells were assayed on day 7 post-sgRNA-transduction. R denotes the Pearson's correlation co-efficient which was calculated for log-transformed values on all genes/promoters that survived filtering except *Cd4*. The average of two biological replicates within a group is shown. Source data are provided as a Source Data file.

surface CD4. In stark contrast, ~80% of the cells containing TETact with the bipartite activator (v3) exhibited surface CD4 (MFI ~ 20,000) from day 4 ($P < 0.001$ vs control, $t$-test)(Fig. 2a–c). In this population significant activation was seen as early as day 2 post-sgRNA transduction, with 50% of cells exhibiting detectable surface CD4 ($P < 0.01$ vs control, $t$-test). Bisulphite sequencing of the *Cd4* promoter in the A20-TETact-v3 cells confirmed the demethylation of the region several hundred base pairs in length (Supplementary Fig. 4a). On the other hand, when tripartite activators (TETact v1 & v2) were recruited, activation was less effective, with these cells showing a lower percentage of CD4 + cells (Fig. 2b) with a lower expression level (Fig. 2c and Supplementary Fig. 3). Of note, by recruiting TET1CD alone (SunTag-TET1), CD4 expression became detectable on d7 to d14 ($P < 0.001$, $t$-test), suggesting DNA methylation indeed plays a role in suppressing CD4 in B cells (Fig. 2b, c and Supplementary Fig. 3).

To evaluate the specificity of our TETact system, we conducted RNA-seq on A20 cells containing the TETact-v3 system targeting the *Cd4* promoter, along with non-targeting control TETact-v3 cells as well as wildtype A20 (Fig. 2d and Supplementary Fig. 4b). Non-transduced A20 showed a similar gene expression profile with the control TETact-v3 cells (Supplementary Fig. 4b, Supplementary Data 1), with differentially expressed genes likely due to lentiviral transformation. Importantly, comparison of TETact-v3 cells targeting *Cd4* promoter with cells expressing non-targeting sgRNA revealed *Cd4* as the sole significantly upregulated gene (Fig. 2d, adjusted $p$-value $= 1.96 \times 10^{-5}$). Expression of the other genes in *Cd4*-targeting sample correlates strongly with the control sample ($R \sim 0.98$).

A previous study has reported that Cas9-TET can lead to off-target DNA demethylation[17]. To examine whether this is also the case for our TETact-v3 system we performed WGBS on non-transduced A20, TETact-v3 control and *Cd4*-targeted TETact-v3 cells. Genome-wide analysis of this high-coverage data (~10x) did not reveal any statistically significant differentially methylated regions (DMRs) between any of the samples suggesting that TETact-v3 does not lead to substantial off-target DNA demethylation at a genome-wide scale. Although the genome-wide approach did not reveal any statistically significant DMRs, an examination specifically focussed on promoter regions (2 kb ± of TSS), revealed changes between the non-transduced A20 cell line and the control TETact lines but no effect on the *Cd4* promoter (Supplementary Fig. 4c and Supplementary Data 2). Importantly, examining the DMRs between the TETact-v3 control and *Cd4*-targeted TETact-v3 revealed a single DMR in the *Cd4* promoter (Fig. 2e and Supplementary Data 2) which correlates with the gene activation observed by RNA-seq (Fig. 2d). Together these experiments reveal that TETact-v3 (henceforth called TETact) system is able to specifically activate silenced genes.

### Demethylation activity is required for TETact-driven gene activation

Combining TET1CD with activators revealed an improved activation of genes with methylated CpG (Figs. 1d and 2), and TETact was shown to demethylate the targeted promoter (Figs. 1e and 2e Supplementary Fig. 4a), however, it still remains uncertain whether the catalytic activity of TET1CD is required to activate gene expression. To address this, we engineered a catalytically dead version of TET1CD into the existing system (DEADTETact). In stark contrast to TETact, DEADTETact resulted in only 6.8% of A20 cells upregulating CD4 (Supplementary Fig. 5a). This strongly suggests that gene activation is dependent on the DNA demethylating activity of TET1CD.

### TETact is effective at activating genes in multiple cell lines

To further validate the TETact system, we attempted to activate CD4 in additional cell lines – 3T3 fibroblasts, MPC11 myeloma and J558L plasmacytoma. Publicly available 3T3 WGBS data[17] revealed that the *Cd4* promoter is heavily methylated in 3T3 (Supplementary Fig. 6a).

Unsurprisingly, SAM failed to activate CD4 in this heavily methylated context (Supplementary Fig. 5b); however, TETact was able to robustly activate CD4 in 3T3 (Supplementary Fig. 5c). Similarly, TETact, but not SAM, successfully activated CD4 in MPC11 and J558L cells (Supplementary Fig. 5d–g). Next, we sought to validate the ability of TETact to activate other genes in the A20 cell line. The T-cell specific receptor genes *Cd3d*, *Cd3e*, *Cd3g* and *Cd8b* are all heavily methylated in B cells, as revealed by the WGBS data (Supplementary Fig. 6b–e). Individually targeting these promoters with TETact in A20 cells significantly activated a higher level of transcription than SAM for all corresponding genes (Fig. 3a, all $P < 0.05$ vs SAM). The activation was accompanied by robust CpG demethylation of several hundred bp at the promoter (Fig. 3b). We also explored a potential application of TETact, in which we attempted to activate embryonic globin genes in 'adult' cells, a major aim of gene therapy to treat hemoglobinopathies[18–20]. Adult haemoglobin is composed of α and β chains and mutations in these genes can lead to various blood disorders, for instance, α- and β-thalassaemia as well as sickle cell anaemia[21]. In contrast, during embryonic development haemoglobin is instead composed of other globin chains and reactivation of these represents a promising therapeutic cure for such disorders[18–20]. WGBS data[13] showed highly methylated DNA across both loci in both B and T cells (Supplementary Fig. 6f, g). We therefore designed sgRNAs to target the murine embryonic α-like ζ-globin (*Hba-x*) and β-like ε_y-globin (*Hbb-y*) in the A20 cell line. qRT-PCR analysis revealed that TETact outperformed the other systems in upregulating *Hba-x* and *Hbb-y* (Supplementary Fig. 7a, b). The activation is also associated with DNA demethylation at the *Hba-x* and *Hbb-y* promoters (Supplementary Fig. 7c, d). In contrast to the A20 cell line, the *Hba-x* and *Hbb-y* promoters have low levels of CpG methylation in 3T3 cells (Supplementary Fig. 6h, i). This is reflected in both SAM and TETact are both capable of activating gene transcription in this cell line. (Supplementary Fig. 7e, f).

### Simultaneous gene activation by TETact

We next tested the ability of the TETact system to simultaneously activate multiple genes in a single cell. Since qRT-PCR is incapable of conveying definitive information at a single cell level, we hence sought to activate multiple surface receptors in A20 cells and assessed the co-expression by flow cytometry. Efficient surface expression of CD8β has long been shown to require co-expression of CD8α chain[22–24], whereas CD8α does not and can exist as CD8αα homodimers on the cell surface[25,26]. In agreement with this, TETact was able to mediate pronounced surface expression of CD8α (Supplementary Fig. 8a middle panel), while targeting *Cd8b* did not result in detectable surface CD8β (Supplementary Fig. 8a right panel), although a robust activation of *Cd8b* was observed at the transcriptional level (Fig. 3a). With this in mind, a lentiviral vector co-expressing both *Cd8a*- and *Cd8b*-promoter targeting sgRNAs was designed and subsequent transduction led to robust expression of CD8α and CD8β (Supplementary Fig. 8b). The existence of a small percentage of CD8α + CD8β- and negligible CD8α-CD8β + cells neatly aligns with the aforementioned dependence of CD8α for efficient CD8β surface expression. Additionally, a marked reduction of CD8α + CD8β + cells was observed in DEADTETact cells, with only 10% of cells expressing both CD8α and CD8β, compared to 92% seen in TETact (Supplementary Fig. 8b). We next attempted to activate CD4, CD8α and CD8β in TETact-expressing A20 cells. Using CD8β as a proxy for CD8α-CD8β co-expression, the activation revealed a 49% of CD4 + CD8β + cells, suggesting a highly efficient co-expression of both CD4, CD8α and CD8β at the surface (Fig. 3c). In contrast, few CD4 + CD8β + cells were observed in DEADTETact cells (Fig. 3c).

## Discussion

The existing second generation CRISPR activators induce transcription through recruitment of various chromatin modifying proteins and

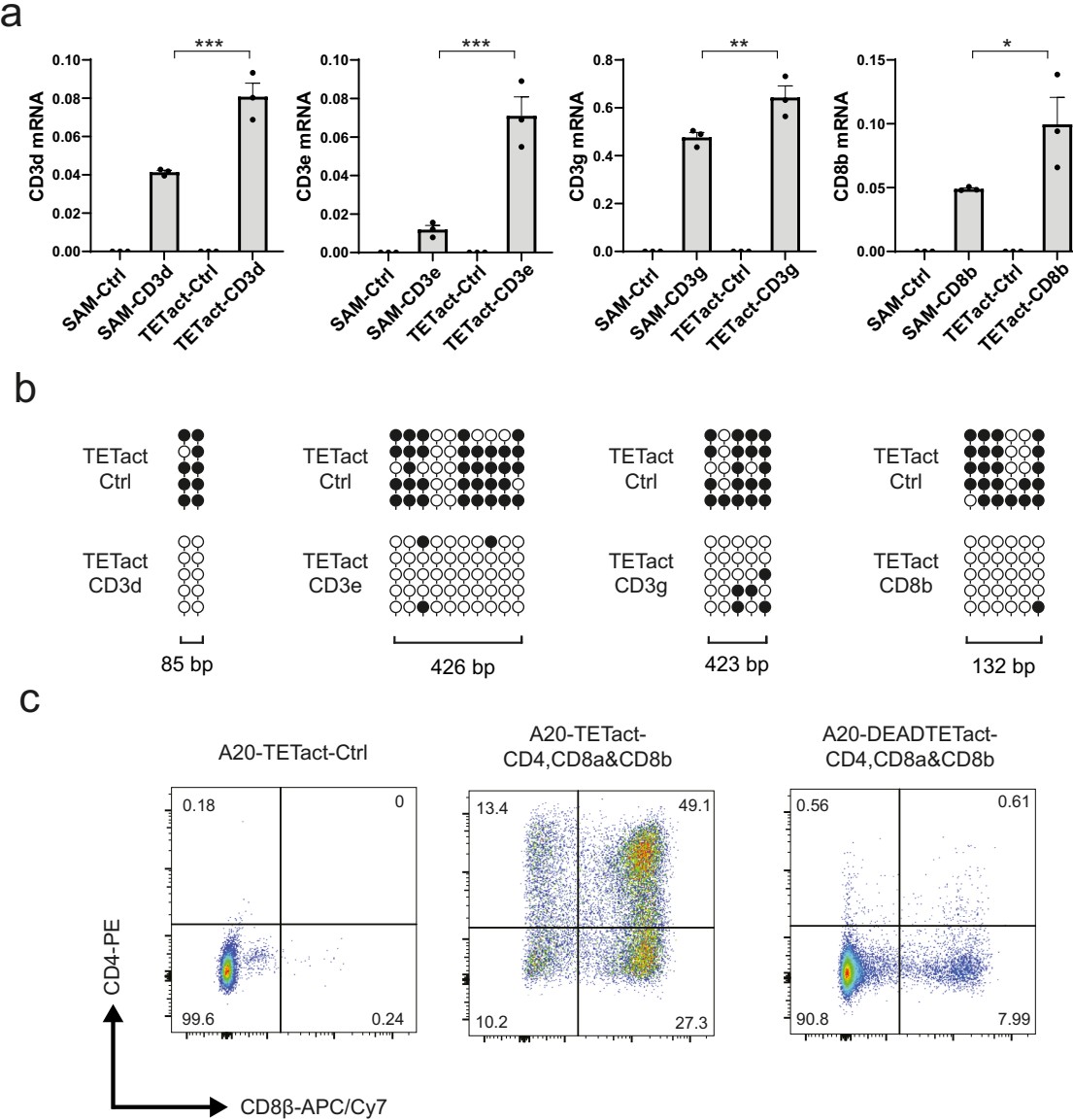

**Fig. 3 | TETact activation of other genes in A20. a** Expression of *Cd3d*, *Cd3e*, *Cd3g* and *Cd8b* in A20 SAM or TETact cells transduced with the promoter-targeting sgRNA. Expression level is relative to *Actb* as $2^{-\Delta Ct}$. $P = 0.0002$ (*Cd3d*), 0.0002 (*Cd3e*), 0.0091 (*Cd3g*), 0.0391 (*Cd8b*) from one-way ANOVA with Tukey's *post hoc* test. Data are shown as mean ± s.e.m. from three independent transductions. *$P < 0.05$, **$P < 0.01$, ***$P < 0.001$. **b** Bisulphite sequencing of *Cd3d*, *Cd3e*, *Cd3g* and *Cd8b* promoters for A20-TETact cells transduced with either control or the corresponding sgRNAs. Open lollipops indicate non-methylated CpG dinucleotides whereas closed lollipops represent methylated CpG motifs. Each row represents an individual clone. Five clones were analysed in each group. **c** Representative flow cytometry plots showing CD4 and CD8β (proxy for CD8αβ co-expression) surface expression in A20-TETact and DEADTETact cells transduced with vector co-expressing *Cd4*-, *Cd8a*- and *Cd8b*-targeting sgRNAs. Cells were assayed on day 7 post-sgRNA-transduction. Gates were drawn based on the negative population of cells transduced with control sgRNA in the same experiment. Source data are provided as a Source Data file.

transcription factors which alter the local epigenetic landscape such as histone modifications and nucleosome spacing[6,8]. However, we found that these systems were inefficient at activating genes that contained high levels of methylated CpG dinucleotides. Here we demonstrated that simultaneous recruitment of DNA demethylating enzymes and activation domains can lead to a more robust transcriptional activation of stably silenced genes. A co-recruitment system has been described in which the TET1CD and activators competitively bind to the same SunTag GCN4 epitope[27]. We believe that separate tethering of co-factors through different scaffolding partners in a non-competitive manner can maximise both activities. Coincidentally, a similar non-competitive system CRISPRon was developed during the preparation of this manuscript, with direct fusion of a single TET1CD to dCas9 and recruitment of VPR through an RNA scaffold[15]. Whilst this system was able to reverse the repressive state rendered by CRISPR-mediated stable silencing, it is yet to be demonstrated to be able to activate stably and naturally silenced genes. We suspect that the multiple copies of TET1CD recruited by the SunTag epitope in our TETact system are required for robust and efficient gene activation in these settings. The utilisation of SunTag also enables gene delivery via lentivirus, which would be more favourable in certain biological contexts.

Importantly, the activation of stably silenced genes has many important applications from studies of fundamental biology through to gene-editing therapeutics and cellular reprogramming. The robust gene activation and the capability of multiplexing from the TETact system presented here will facilitate these applications.

## Methods

### Cell culture

The A20 cell-line (ATCC, #TIB-208) was cultured in RPMI 1640 with 2 mM GlutaMAX (Invitrogen, #35050061), 50 µM β-mercaptoethanol (Sigma, #M3148) and 10% heat-inactivated foetal calf serum (FCS, Bovogen, #SFBS-AU). MPC11 (ATCC, #CCL-167) and J558L (ECACC, #88032902) cells were cultured in RPMI 1640 with 2 mM GlutaMAX, 50 µM β-mercaptoethanol, 1X Non-essential amino acids (Invitrogen, #11140050) and 10% heat-inactivated FCS. HEK 293 T (ATCC, #CRL-3216) and NIH/3T3 cells (ATCC, #CRL-1658) were cultured in DMEM with 2 mM GlutaMAX and 10% heat-inactivated FCS without antibiotics.

### Plasmid design and construction

The lentiviral vector dCas9-5xGCN4-P2A-BFP was constructed by amplifying the GCN4 array from pCAG-dCas9-5xPlat2AflD (Addgene #82560) with primers bearing the BamHI and NotI sites at the 5' and 3' end respectively, and cloning into the corresponding site in pHRdSV40-dCas9-10xGCN4-P2A-BFP (Addgene #60903). Plasmid scFv-GCN4-sfGFP-TET1CD was constructed by cloning sfGFP-TET1CD fragments from pCAG-scFvGCN4sfGFPTET1CD (Addgene #82561) with BamHI and NotI cuts to the corresponding sites in pHRdSV40-scFv-GCN4-sfGFP-VP64-GB1-NLS (#60904). MCP-p65-hsf1-T2A-mCherry was constructed from Addgene plasmid MS2-P65-HSF1_GFP (#61423) by replacing GFP with an mCherry gene. Vector gRNA-MS2x2-TagRFP657 was constructed from pLH-sgRNA1-2XMS2 (Addgene #75389) by removing the ccdB and replacing with a shorter BbsI cloning cassette, made from annealing complementary oligos, to the BbsI site in the plasmid, an XbaI site was then added upstream to the EcoRI site via PCR, hygromycin resistance gene was further replaced with a TagRFP657 gene obtained from pMSCVpuro-TagRFP657 (Addgene #96939). Based on MCP-p65-hsf1-T2A-mCherry, plasmids MCP-VP64-p65-hsf1-T2A-mCherry and MCP-VPR-T2A-mCherry were constructed via In-Fusion Cloning (Clontech, #638947) with VP64 or VPR obtained from the Addgene plasmid #84244.

The plasmid containing the catalytically dead TET1CD, scFv-GCN4-sfGFP-deadTET1CD, was constructed by using mutagenic primers creating H1672Y and D1674A of TET1 and then assembling into the vector backbone at BamHI and NotI sites via NEBuilder HiFi DNA assembly (NEB, #E5520S).

The related TETact (v3) plasmids have been deposited onto Addgene database (#184438–184442).

For SAM activation, vector dCas9-VP64-mCherry was modified from Addgene plasmid dCas9-VP64-GFP (#61422) by exploiting NheI and EcoRI sites to replace the GFP with an mCherry gene. MCP-p65-hsf1-BFP was modified from Addgene plasmid MS2-P65-HSF1_GFP (#61423) by replacing the GFP with a TagBFP gene. SunTag-VP64 plasmids are the Addgene plasmids #60903 and #60904 described above. Primers are listed in Supplementary Table 1.

Target sites for dCas9 were designed through the IDT online design tool (https://www.idtdna.com/SciTools). For cloning target sequence into the corresponding guide RNA vector, protospacer sequence of 20 bp (Supplementary Table 2) was ordered as a pair of complementary oligos with 4 additional nucleotides ACCG- and AAAC- at the 5' end of the sense and antisense oligonucleotides, respectively. Complementary oligos were annealed by heating at 95 °C for 5 min and subsequent cooling to 22 °C at a rate of −0.1 °C/s. The annealed oligos were then ligated to the BbsI cut site of the vector.

For cloning multiplex sgRNA plasmids, a vector with the first desired sgRNA was digested with XbaI and EcoRI, whereas the entire U6-sgRNA-MS2 cassette for the second and/or third desired sgRNA was amplified by PCR, with the amplicon ends being able to get digested by BbsI to liberate compatible 4-bp overhangs to the adjacent fragments. Ligation was performed by incubating the DNA fragments in the presence of BbsI-HF (NEB, #R3539S) and T4 DNA ligase (NEB, #M0202S) with 60 alternating cycles between 37 °C for 5 min and 16 °C for 5 min.

### Lentivirus production and transduction

One day prior to transfection, HEK293T cells were seeded at a density of $1.2 \times 10^6$ cells/well in a 6-well plate in 2 ml Opti-MEM (Invitrogen, #31985062) containing 2 mM GlutaMAX, 1 mM Sodium Pyruvate (Invitrogen, #11360070) and 5% FCS. Transfection of HEK293T was performed using Lipofectamine 3000 (Invitrogen, #L3000008) as per the manufacturers' instructions. Cells were co-transfected with packaging plasmids (pCMV-VSV-g and psPAX2) at 0.17 pmol each and around 0.23 pmol transfer construct to make up a final mass of 3.3 µg. Virus was harvested 24- and 52-h post-transfection. Transduction was performed in a 12-well plate, with 500,000 cells resuspended in 1 ml viral supernatant supplemented with 8 µg/ml polybrene (Millipore, #TR-1003-G). Cells were spun at 2500 rpm at 32 °C for 90 min. Stable transfectants were enriched by FACS and assayed at the indicated time point, or subjected to further transduction if required.

### Flow cytometry and fluorescence-activated cell sorting (FACS)

For surface marker studies (CD4, CD8α and/or CD8β), cells were assayed at the indicated time point post gRNA transduction. Cells were stained with either CD4-PE (clone GK1.5, in-house, 1:800) or CD8a-PE (clone 53-6.7, BioLegend, #100707, 1:800) or CD8b-APC/Cy7 (clone YTS156.7.7, BioLegend, #126619, 1:600) and analysed with BD FAC-Symphony A3 or BD LSRFortessa and subsequently using FlowJo 10.4.1. For *Dreg1*, *Cd3e*, *Cd3d*, *Cd3g*, *Cd8b*, *Hba-x* and *Hbb-y* studies, cells were sorted on BD FACSAria Fusion or FACSAria III 7 days post gRNA transduction. SAM cells were sorted as mCherry+ BFP + TagRFP657+ population. SunTag-VP64 or SunTag-TET1 cells were sorted as BFP + GFP + TagRFP657 + population. TETact v1-v3 cells were sorted as BFP + GFP + mCherry + TagRFP657 + population. Gating strategies are shown in Supplementary Fig. 9.

### Bisulphite sequencing

Genomic DNA was extracted from around 700,000 cells using DNeasy Blood & Tissue kit (Qiagen, #69506). 200–600 ng of gDNA was then subjected to bisulphite conversion and subsequent clean-up using EpiMark Bisulfite Conversion Kit (NEB, #E3318S) as per manufacturers' instruction. Bisulphite PCR primers for target promoters were designed via Bisulfite Primer Seeker (Zymo, https://www.zymoresearch.com/pages/bisulfite-primer-seeker) and sequences are listed in Supplementary Table 3. Bisulphite PCR was performed using Phusion U Hot Start DNA polymerase (Thermo Fisher, # F555S) or Platinum II Hot Start Taq (Thermo Fisher, #14966001) with resultant amplicon gel purified and cloned into pJET1.2 blunt vector (Thermo Fisher) of the CloneJET PCR cloning kit (Thermo Fisher, # K1231). Five to Ten clones from each group were analysed via Sanger sequencing and subsequently using SnapGene 5.1.0.

### Quantitative reverse transcription PCR (RT-qPCR)

RNA was extracted using NucleoSpin RNA Plus (Macherey-Nagel, #740984) with gDNA removal. One step RT-qPCR was performed in either Bio-Rad CFX384 or QuantStudio 6 Flex using 20 ng RNA with iTaq Universal probe supermix (Bio-Rad, #172-5141) for *Dreg1*, or iTaq Universal Sybr Green supermix (Bio-Rad, #172-5150) for *Cd3e*, *Cd3d*, *Cd3g*, *Cd8b*, *Hba-x* and *Hbb-y*, with β-actin as the endogenous reference. Gene expression was normalised to the endogenous control as ΔCT and relative expression evaluated as $2^{-\Delta CT}$. Primers and probes are listed in Supplementary Table 4.

## RNA-seq

RNA was extracted with NucleoSpin RNA Plus (Macherey-Nagel, #740984) with gDNA removal. Library preparation was performed according to the Illumina TruSeq RNA (100 ng plus, #RS-122-2001) v1.0 protocol. Libraries were sequenced on a NextSeq2000 as 66 bp paired-end reads.

## RNA-seq data analysis

RNA sequencing reads were aligned to the mm10 genome using Rsubread v2.8.1 align[28] and using Rsubread's inbuilt mm10 RefSeq gene annotation. Read counts were obtained for Entrez Gene IDs using featureCounts and Rsubread's inbuild annotation. Gene annotation was downloaded from ftp://ftp.ncbi.nlm.nih.gov/gene/DATA/GENE_INFO (July 2021).

Differential expression analyses were undertaken using the edgeR v3.36.0[29] and limma v3.50.0[30] software packages. Genes without symbols or with duplicated symbols were removed. Unexpressed genes were filtered using edgeR's filterByExpr function with default arguments. Mitochondrial genes, ribosomal RNA genes and genes of type "other" were also filtered. Library sizes were normalized by edgeR's TMM method[31].

Differential expression was assessed using the voom-lmFit approach with the function voomLmFit[32]. This function is an extension of the limma-voom pipeline that takes better account of zero counts[33]. The function transforms the counts to the $\log_2$CPM scale, computes voom precision weights and fits limma linear models. This was followed by applying robust empirical Bayes to the fitted model[34]. The design matrix was constructed using a layout that specified the group. $P$-values were adjusted using the Benjamini and Hochberg method. Significance is defined using an adjusted $p$-value cutoff that is set at 5%.

The cpmByGroup function was used to calculate the average expression ($\log_2$CPM) for all genes that survived filtering. The Pearson's correlation co-efficient between groups was calculated using the average expression ($\log_2$CPM) of all filtered genes except the targeted gene *Cd4*.

## Whole genome Enzymatic Methyl-seq (EM-seq)

Genomic DNA was extracted from around 700,000 cells using DNeasy Blood & Tissue kit (Qiagen, #69506) and 200 ng of gDNA was sheared into size of around 240–290 bp. Libraries were prepared from the sheared gDNA using the NEBNext Enzymatic Methyl-seq kit (NEB, #E7120S) as per manufacturers' instruction. Libraries were sequenced on a NextSeq2000 as 66 bp paired-end reads for 100 cycles.

## Whole genome methylation analysis

Adaptors and reads of poor quality scores were trimmed using Trim-Galore v0.6.7[35] with default settings. Paired-end EM-seq reads were aligned to the mouse mm10 genome using bismark v0.20.0[36]. Duplicated reads were removed, and methylation calling were also performed in bismark with default settings.

Differential methylation analyses were performed using the bsseq v1.32.0[37], DMRcate v2.10.0[38,39] and edgeR v3.38.1[29] software packages. Methylated and unmethylated CpG reads counts of bismark methylation coverage outputs were smoothed across the genome using bsseq's BSmooth function with default settings. Differential methylation was assessed for all CpG loci across the genome using DMRcate's sequencing.annotate function with default parameters.

Gene promoter methylation signal for each gene was obtained by aggregated the methylated and unmethylated CpG read counts in the region from 2 kb upstream to 2 kb downstream of the transcription start site (TSS) of that gene. Gene promoters with at least 10 CpG coverage in all the samples were kept in the analysis. Differentially methylation in gene promoter regions was assessed following the edgeR differentially methylation pipeline[40]. DNA methylation level ($M$-value) of promoter was calculated using the $\log_2$ ratio of the methylated versus unmethylated reads.

## Statistics and reproducibility

All data analyses were performed with GraphPad Prism 9. Data were expressed as mean ± s.e.m. For comparison of three or more experimental conditions, one-way ANOVA was used followed by Dunnett's or Tukey's *post hoc* analysis. Two-tailed student's $t$-test was used for comparison of two experimental conditions. Comparisons with $P < 0.05$ were considered statistically significant. In vitro experiments were repeated at least three times independently with similar results obtained unless otherwise stated.

For RNA-seq, a moderated $t$-test was performed for each gene. For EM-seq, a quasi-likelihood $F$-test was performed for each gene promoter. All the $P$-values are two-sided, and the BH (Benjamini and Hochberg) method was used for multiple comparisons $P$-values adjustment. RNA-seq and EM-seq libraries were prepared from two independently repeated samples.

## Reporting summary

Further information on research design is available in the Nature Research Reporting Summary linked to this article.

## Data availability

The RNA-seq and EM-seq data generated in this study are tabulated as supplementary data 1 & 2 respectively, and have been deposited in the GEO database under accession numbers GSE203162 and GSE211754, under the SuperSeries GSE212345. WGBS data of naïve B and T cells was retrieved from GSE94674, whereas that of 3T3 cells was retrieved from GSE162138. All other relevant data are available within the article and its Supplementary Information files. Source data are provided with this paper.

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

## Acknowledgements

We thank the staff of the core facilities at the Walter and Eliza Hall Institute. This work was supported by grants and fellowships from the Marian and E.H. Flack Fellowship (H.D.C.), the National Health and Medical Research Council of Australia (C.R.K #1125436, T.M.J. #1124081, R.S.A. #1100451, G.K.S. & R.S.A. #1158531), Medical Research Future Fund (MRFF) Investigator Grant (Y.C. #1176199) and the Australian Research Council (R.S.A. #130100541). This study was made possible through Victorian State Government Operational Infrastructure Support, the Australian Government NHMRC Independent Research Institute Infrastructure Support scheme, and the Australian Cancer Research Fund.

## Author contributions

W.F.C. designed and conducted experiments, T.M.J. conducted RNA-seq experiment. H.D.C. and Y.C. performed the bioinformatic analysis. C.R.K. provided intellectual input. G.K.S., A.C.P., T.M.J. and R.S.A. oversaw the study. W.F.C. and R.S.A. conceived the study and wrote the manuscript.

## Competing interests

The authors declare no competing interests.
