## [Peer Review File · Nature Communications]

Reviewers' Comments:

Reviewer #1:

Remarks to the Author:

NCOMMS-21-34545-T

Activation of stably silenced genes by recruitment of a synthetic de-methylating module
Wing Fuk Chan et al.

Highly methylated genes are stably silenced and resistant to dCas9-based activation systems. To counter this, Wing Fuk Chan et al. reported an activation system coupling the catalytic domain of DNA demethylating enzyme TET1 with transcriptional activators (TETact). This system adopted the previously described SunTag approach for the recruitment of TET1CD. In addition, this system adopted the previously described RNA aptamer MS2 (SAM) within the sgRNA for the recruitment of different combinations of transcription activators. There are three versions for this system, TETact V1-3, and these versions used different transcription activators. Wing Fuk Chan et al. found TETact V3, which used p65-Hsf1, is the best in these versions. TETact V3 showed significant activation compared to V1, V2, SAM, SunTag recruiting TET1, and SunTag recruiting VP64. However, an activation system coupling the catalytic domain of DNA demethylating enzyme TET1 with transcriptional activators, was previously reported. In the previous reported system, both TET1 and transcriptional activators were presented on SunTag. Specially, the combination of TET1 and p65-Hsf1 showed significant synergetic effect as reported in the present manuscript. In light of these, we do not think there is much new in this paper.

Comments

1. The authors ignored a previously reported paper (Int J Mol Sci. 2020 Feb 25;21(5):1574.) which described an activation system coupling the catalytic domain of DNA demethylating enzyme TET1 with transcriptional activators such as p65-Hsf1.
2. Only one stable cell line (A20) was used for validation of the system in this study. It is important to examine multiple lines for each experiment because a stable cell line often shows biased epigenome.
3. Only three genes were used for validation of the system. To generalize the results, systematic experiments are preferred.
4. Validation of off-targets effect is also important in epigenome editing. The authors should perform these experiments.

Reviewer #2:

Remarks to the Author:

The manuscript presents an epigenetic mechanism to explain the failure of current dCas9 gene regulation systems to activate certain targets. It also provides an improved dCas9 effector to better activate epigenetically silenced genes that cannot be activated by classical dCas9 activators. Both the described mechanism of CRISPRa failure and produced technology are interesting and valuable from a biological, as well as technical standpoint. The preliminary evidence provided in this manuscript illustrates a potential correlation between CpG methylation and inability of being activated by current-day dCas9 modulators. The authors subsequently demonstrated a new CRISPRa platform, TETact, to activate select methylated genes which proved resistant to modulation by classical dCas9 effectors. The experimental design, methods and conclusions of this research are reasonable and well-written. However, the paper requires significant revisions to sufficiently support the conclusions, in order to merit publication in Nature Communications.

Major revisions

Further validation of tool function

The authors didn't provide enough evidence to show that the enhanced activation ability of

TET1CD is due to its demethylation activity. Further proof that TET1CD is directly responsible for demethylating CpG sites and that this is the critical function it plays in their TETact system is lacking. Showing that a catalytically dead version of TETact fails to activate gene expression and fails to demethylate the target locus would strengthen the story by proving that the catalytic activity of TET1CD is necessary for their observed effects (and not a scaffolding function of the protein instead). Showing that the other dCas9 activators that fail to activate the target genes tested do not demethylate the target loci using bisulfite sequencing would also lend support to the claim that TET1CD-driven demethylation is necessary for gene activation of methylated loci.

Validation in multiple cell lines and genes

First, to better demonstrate that the correlation of CpG methylation and inability of being activated is not a cell line- or gene- specific effect, the authors should repeat the same activation experiments in multiple cell lines. Ideally the authors could include cell lines with different CpG methylation statuses of the same gene which will allow them to show that in cell lines where the locus is not methylated conventional dCas9 activators work, but in other cells where the locus is methylated the TETact system is required. Furthermore, too few gene targets were tested within the preliminary studies presented and thus it is unclear how significant their findings are. Additional targets should need to be examined. Second, targeted bisulfite sequencing data of all the targeted genes before and after their treatment should be provided. These experiments will help more firmly establish their findings and also demonstrate more conclusively the utility of their TETact tool.

Analysis of off-target effects

The comparison of control (non-targeting) guide vs CD4 targeting guide appears to show that the recruitment of TET1CD to the CD4 locus leads to a large amount of demethylation. However, experiments with other dCas9 epigenetic modifiers have shown them to be rather promiscuous (e.g. dCas9-DMNT3a and dCas9-p300 fusions), and for this reason a more global analysis of specificity is necessary. In addition, as these tools have been shown to induce off-target effects independent of the gRNA provided (e.g. dCas9-p300), these analyses need to include comparisons between non-transfected cells, and cells transfected with TETact with a targeting gRNA against a locus of interest and a control gRNA (3 total experimental groups being compared). Furthermore, if significant amounts of unexpected demethylation are seen across the genome in cells that receive TETact versus non-transfected cells, RNAseq should be provided to put those findings into context. In addition, if significant global off-target activity from TETact is seen experiments should be performed to demonstrate that the off-target demethylation TETact performs is not sufficient to enable a dCas9 activator to induce the expression of silenced loci (i.e. that you need both the TET1CD and the activator to be colocalized to the locus of interest to see gene activation).

Sequences of developed tools

At a minimum all tools described in this manuscript should have their full sequences provided as part of the supplement in order to enable other users to replicate these finding and adopt their method. In addition, it would be ideal if all reagents were deposited in a non-profit plasmid repository such as Addgene to facilitate their dissemination.

Inconsistency between replicates

Figure 2A shows a very strong phenotype in which CD4 is highly activated at d14 by TETact-v3. In this figure, activated cells form one highly concentrated population that is highly CD4-PE and mCherry positive. Supplementary figure 3 appears to show a much weaker phenotype in which CD4 is less strongly activated at d14 by TETact-v3. In this case, the mCherry positive cells form a continuum of slightly CD4-PE positive cells. This difference should be explained.

Minor revisions/points

Multiple target activation by TETact

The ability of TETact to activate multiple genes at once is something commonly performed in the activation literature and would strengthen the utility of this tool. The authors can easily test multiplexed targeting by introducing TETact into cells along with their existing set of sgRNAs.

Elaboration of data analysis

Figure 2C shows the flow data of gene activation in a time-dependent manner. However, in the flow plot (Figure 2A) of d7 and d14, the median CD4 expression level in non-activated cells in the population is higher than in ctrl and d2, rendering more cells in the positive gate. The flow experiments and data analysis (e.g. normalization of MFI) should be further elaborated to explain the discrepancy among different time points and how they were controlled for.

RESPONBSE TO REVIEWER COMMENTS NCOMMS-21-34545A

Reviewer #1 (Remarks to the Author):

NCOMMS-21-34545-T

Activation of stably silenced genes by recruitment of a synthetic de-methylating module

Wing Fuk Chan et al.

Highly methylated genes are stably silenced and resistant to dCas9-based activation systems. To counter this, Wing Fuk Chan et al. reported an activation system coupling the catalytic domain of DNA demethylating enzyme TET1 with transcriptional activators (TETact). This system adopted the previously described SunTag approach for the recruitment of TET1CD. In addition, this system adopted the previously described RNA aptamer MS2 (SAM) within the sgRNA for the recruitment of different combinations of transcription activators. There are three versions for this system, TETact V1-3, and these versions used different transcription activators. Wing Fuk Chan et al. found TETact V3, which used p65-Hsf1, is the best in these versions. TETact V3 showed significant activation compared to V1, V2, SAM, SunTag recruiting TET1, and SunTag recruiting VP64. However, an activation system coupling the catalytic domain of DNA demethylating enzyme TET1 with transcriptional activators, was previously reported. In the previous reported system, both TET1 and transcriptional activators were presented on SunTag. Specially, the combination of TET1 and p65-Hsf1 showed significant synergetic effect as reported in the present manuscript. In light of these, we do not think there is much new in this paper.

Comments

1. The authors ignored a previously reported paper (Int J Mol Sci. 2020 Feb 25;21(5):1574.) which described an activation system coupling the catalytic domain of DNA demethylating enzyme TET1 with transcriptional activators such as p65-Hsf1.

Reply:

Many thanks to reviewer for reminding us about this existing relevant reference, and we apologise for the overlook of this important paper. We have now cited and discussed this paper in our discussion section.

2. Only one stable cell line (A20) was used for validation of the system in this study. It is important to examine multiple lines for each experiment because a stable cell line often shows biased epigenome.

Reply:

We agree with the reviewer. As suggested, we have repeated the activation in the fibroblast cell line 3T3, the myeloma MPC11 and the plasmacytoma cell line J558L (Supplementary Fig. 4).

3. Only three genes were used for validation of the system. To generalize the results, systematic experiments are preferred.

Reply:

We thank reviewer for the comment and recommendation. In addition to the Dreg1, Cd4, Hba-x and Hbb-y (Fig.1, 2, Supplementary Fig. 6), we then also activated more genes – Cd3d, Cd3e, Cd3g, Cd8a and Cd8b (Fig. 3)

4. Validation of off-targets effect is also important in epigenome editing. The authors should perform these experiments.

Reply:

We appreciate the suggestion from reviewer. We have thus conducted an RNA-seq experiment on the WT A20, the TETact cells with control guide and the TETact cells with CD4 targeting sgRNA (Fig. 2e). These analyses suggested that the gene expression is highly similar ($R \sim 0.98$) except for the target-activated gene – Cd4.

Reviewer #2 (Remarks to the Author):

The manuscript presents an epigenetic mechanism to explain the failure of current dCas9 gene regulation systems to activate certain targets. It also provides an improved dCas9 effector to better activate epigenetically silenced genes that cannot be activated by classical dCas9 activators. Both the described mechanism of CRISPRa failure and produced technology are interesting and valuable from a biological, as well as technical standpoint. The preliminary evidence provided in this manuscript illustrates a potential correlation between CpG methylation and inability of being activated by current-day dCas9 modulators. The authors subsequently demonstrated a new CRISPRa platform, TETact, to activate select methylated genes which proved resistant to modulation by classical dCas9 effectors. The experimental design, methods and conclusions of this research are reasonable and well-written. However, the paper requires significant revisions to sufficiently support the conclusions, in order to merit publication in Nature Communications.

Major revisions

Further validation of tool function

The authors didn't provide enough evidence to show that the enhanced activation ability of TET1CD is due to its demethylation activity. Further proof that TET1CD is directly responsible for demethylating CpG sites and that this is the critical function it plays in their TETact system is lacking. Showing that a catalytically dead version of TETact fails to activate gene expression and fails to demethylate the target locus would strengthen the story by proving that the catalytic activity of TET1CD is necessary for their observed effects (and not a scaffolding function of the protein instead). Showing that the other dCas9 activators that fail to activate the target genes tested do not demethylate the target loci using bisulfite sequencing would also lend support to the claim that TET1CD-driven demethylation is necessary for gene activation of methylated loci.

Reply:

We agree with the reviewer that the evidence presented in the original manuscript is insufficient regarding the mechanism behind the TETact mediated activation, and we appreciate this suggestion. We therefore constructed the catalytically dead TET1CD version for the system (DEADTETact) and showed that the CD4 activation was greatly impaired (Supplementary Fig. 4a). We also showed that

DEADTETact is, in contrast to TETact, unable to efficiently activate CD4, CD8a and CD8b in a multiplexing context (Fig. 3c, Supplementary Fig. 7). Overall the new data with the DEADTETact suggested the catalytic, and therefore the DNA demethylating, activity of TET1CD is responsible for the observed TETact-mediated activation.

Validation in multiple cell lines and genes

First, to better demonstrate that the correlation of CpG methylation and inability of being activated is not a cell line- or gene- specific effect, the authors should repeat the same activation experiments in multiple cell lines. Ideally the authors could include cell lines with different CpG methylation statuses of the same gene which will allow them to show that in cell lines where the locus is not methylated conventional dCas9 activators work, but in other cells where the locus is methylated the TETact system is required. Furthermore, too few gene targets were tested within the preliminary studies presented and thus it is unclear how significant their findings are. Additional targets should need to be examined. Second, targeted bisulfite sequencing data of all the targeted genes before and after their treatment should be provided. These experiments will help more firmly establish their findings and also demonstrate more conclusively the utility of their TETact tool.

Reply:

We totally agree with the reviewer. As suggested, we have repeated the activation in the fibroblast cell line 3T3, the myeloma MPC11 and the plasmacytoma cell line J558L (Supplementary Fig. 4b-e). In addition to the Dreg1, Cd4, Hba-x and Hbb-y (Fig.1, 2, Supplementary Fig. 6), we also activated more genes – Cd3d, Cd3e, Cd3g, Cd8a and Cd8b (Fig. 3). Targeted bisulphite sequencing of all the methylated target genes in TETact cells with control vs promoter-targeting sgRNA are also provided (Fig. 1e, 3b, Supplementary Fig. 6c, d). Lastly, we also compared the efficiency of SAM and TETact in activating unmethylated genes (Supplementary Fig. 6) – same genes (Hba-x, Hbb-y) in a different cell lines (3T3).

Analysis of off-target effects

The comparison of control (non-targeting) guide vs CD4 targeting guide appears to show that the recruitment of TET1CD to the CD4 locus leads to a large amount of demethylation. However, experiments with other dCas9 epigenetic modifiers have shown them to be rather promiscuous (e.g. dCas9-DMNT3a and dCas9-p300 fusions), and for this reason a more global analysis of specificity is necessary. In addition, as these tools have been shown to induce off-target effects independent of the gRNA provided (e.g. dCas9-p300), these analyses need to include comparisons between non-transfected cells, and cells transfected with TETact with a targeting gRNA against a locus of interest and a control gRNA (3 total experimental groups being compared). Furthermore, if significant amounts of unexpected demethylation are seen across the genome in cells that receive TETact versus non-transfected cells, RNAseq should be provided to put those findings into context. In addition, if significant global off-target activity from TETact is seen experiments should be performed to demonstrate that the off-target demethylation TETact performs is not sufficient to enable a dCas9 activator to induce the expression of silenced loci (i.e. that you need both the TET1CD and the activator to be colocalized to the locus of interest to see gene activation).

Reply:

We thank reviewer for the suggestion. We have therefore performed RNA-seq experiment on the WT A20, the TETact cells with control guide and the TETact cells with CD4 targeting sgRNA (Fig. 2e). The gene expression is highly similar ($R \sim 0.98$) except for the activated gene – Cd4.

Sequences of developed tools

At a minimum all tools described in this manuscript should have their full sequences provided as part of the supplement in order to enable other users to replicate these finding and adopt their method. In addition, it would be ideal if all reagents were deposited in a non-profit plasmid repository such as Addgene to facilitate their dissemination.

Reply:

As suggested we have now deposited the corresponding plasmids – dCas9-5xGCN4-P2A-BFP; scFv-GCN4-sfGFP-TET1CD; MCP-p65-hsf1-T2A-mCherry; gRNA-MS2x2-TagRFP657 and scFv-GCN4-sfGFP-deadTET1CD – to the Addgene. The Addgene plasmid numbers were assigned (#184438-184442) and outlined in the method section of the manuscript. They will become publicly available once this study is published.

Inconsistency between replicates

Figure 2A shows a very strong phenotype in which CD4 is highly activated at d14 by TETact-v3. In this figure, activated cells form one highly concentrated population that is highly CD4-PE and mCherry positive. Supplementary figure 3 appears to show a much weaker phenotype in which CD4 is less strongly activated at d14 by TETact-v3. In this case, the mCherry positive cells form a continuum of slightly CD4-PE positive cells. This difference should be explained.

Reply:

Many thanks to reviewer for attention to details, and we apologise for having the typo in this Supplementary figure 3 – the TETact version designation was mixed up. There were not any plots of -v3 in the supplementary figure 3 but instead plots for -v1 (VPR) and -v2 (VP64-p65-hsf1). We have now corrected the typo in the figure as well as the figure legend accordingly.

Minor revisions/points

Multiple target activation by TETact

The ability of TETact to activate multiple genes at once is something commonly performed in the activation literature and would strengthen the utility of this tool. The authors can easily test multiplexed targeting by introducing TETact into cells along with their existing set of sgRNAs.

Reply:

We greatly appreciate this suggestion and therefore performed these multiplex experiments. We activated 2 surface receptors – CD8 α and CD8 β simultaneously in A20 (Supplementary Fig. 7b); then we also activated 3 surface receptors – CD4, CD8 α and CD8 β – at once in A20 (Fig. 3c).

Elaboration of data analysis

Figure 2C shows the flow data of gene activation in a time-dependent manner. However, in the flow plot (Figure 2A) of d7 and d14, the median CD4 expression level in non-activated cells in the population is higher than in ctrl and d2, rendering more cells in the positive gate. The flow experiments and data analysis (e.g. normalization of MFI) should be further elaborated to explain the discrepancy among different time points and how they were controlled for.

Reply:

We have now added the gate setting strategies in each of the flow cytometry data figure legends. In brief, all the gates were set using the negative population of the control sgRNA transduced cells examined at each time point of the experiment.

Reviewers' Comments:

Reviewer #1:

Remarks to the Author:

The comments are very well addressed except for the following points.

Chan et al described as follows in the discussion.

"A synergistic system has been described in which the TET1CD and activators competitively bind to the SunTag GCN4 epitope²⁷. The recruitment of one factor is at the expense of binding of another factor, as a result both the CpG demethylating and gene activating activities will be compromised and are thus sub-optimal."

This is entirely the authors' speculation and not based on any experimental evidence. It is possible that having the two factors in close proximity on the SunTag may produce a better effect than presenting them separately on the SunTag and SAM. Therefore, The authors need to do experiments to prove which is more effective, presenting TET1 and p65-hsf1 on the same SunTag or presenting them separately on SunTag and SAM.

Reviewer #2:

Remarks to the Author:

Overall the authors addressed many of my initial comments. Two points remain from my initial comments that were not fully addressed:

1) For experiments where the author's show that TETact enables the activation of methylated genes it is important that the authors always include the control where they use an activator that doesn't recruit TET1. For example for Figure 3b and Supplementary Figure 4d&4e they show that TETact can induce the expression of their target genes but critical to their thesis is that a normal activator that doesn't recruit TET1 cannot activate these genes, yet they do not show these data. The reviewers should show these controls as it is essential to include them to help support their hypothesis.

2) The authors need to perform whole genome bisulfite sequencing in their TetAct with gRNA against target gene (e.g. CD4) vs TetAct with gRNA against control gene (e.g. luciferase) vs control line with no TetAct to demonstrate specificity of their system. The authors make statements such as "Together these experiments reveal the exquisite specificity of the TETactv3 (henceforth called TETact) system", but as I tried to point out in my initial comment to the authors these epigenetic tools have been repeatedly found to have off target effects on the epigenome even if these do not lead to transcriptional changes that can be observed by RNA-sequencing. Furthermore, other papers using Tet1 dCas9 fusions have found that independent of the gRNA when expressed in cells these tools can lead to the loss of methylation from loci across the genome (including the locus they wish to target). Thus, I would strongly encourage the authors to perform the above WGBS experiment with all 3 conditions and report their findings. Given the previously published data (references provided in my previous reviews plus see article in Nature Communications 2021 from Sapozhnikov and Szyf; <https://doi.org/10.1038/s41467-021-25991-9>) I anticipate that the authors will find that their TETact system causes off target epigenome modifications as compared to the control cells without TETact, but it is critical that they report this in their paper as others who are not experts in the field and are naïve to the nuance of these tools will not be aware of this caveat and thus will not be aware of its limitations.

REVIEWER COMMENTS

Reviewer #1 (Remarks to the Author):

The comments are very well addressed except for the following points.

Chan et al described as follows in the discussion.

“A synergistic system has been described in which the TET1CD and activators competitively bind to the SunTag GCN4 epitope²⁷. The recruitment of one factor is at the expense of binding of another factor, as a result both the CpG demethylating and gene activating activities will be compromised and are thus sub-optimal.”

This is entirely the authors' speculation and not based on any experimental evidence. It is possible that having the two factors in close proximity on the SunTag may produce a better effect than presenting them separately on the SunTag and SAM. Therefore, The authors need to do experiments to prove which is more effective, presenting TET1 and p65-hsf1 on the same SunTag or presenting them separately on SunTag and SAM.

Reply:

Many thanks to reviewer for the appreciation of our efforts in addressing the previous comments. We admit that we have made some very bold speculation about the mentioned system, and we have thus removed those lines in the manuscript.

Reviewer #2 (Remarks to the Author):

Overall the authors addressed many of my initial comments. Two points remain from my initial comments that were not fully addressed:

1) For experiments where the author's show that TETact enables the activation of methylated genes it is important that the authors always include the control where they use an activator that doesn't recruit TET1. For example for Figure 3b and Supplementary Figure 4d&4e they show that TETact can induce the expression of their target genes but critical to their thesis is that a normal activator that doesn't recruit TET1 cannot activate these genes, yet they do not show these data. The reviewers should show these controls as it is essential to include them to help support their hypothesis.

Reply:

We are grateful that reviewer appreciated our efforts in addressing the initial comments, and we apologised for overlook and not performing the appropriate controls in certain experiments. We have now added the experiments with SAM (no recruitment of TET1) (Fig. 3a, Supplementary Fig. 5d, f), where it showed in general TETact is able to induce a higher level of transcription than SAM for CpG methylated region (CD3e, CD3d, CD3g, CD8b, CD4).

2) The authors need to perform whole genome bisulfite sequencing in their TetAct with gRNA

against target gene (e.g. CD4) vs TetAct with gRNA against control gene (e.g. luciferase) vs control line with no TetAct to demonstrate specificity of their system. The authors make statements such as “Together these experiments reveal the exquisite specificity of the TETactv3 (henceforth called TETact) system”, but as I tried to point out in my initial comment to the authors these epigenetic tools have been repeatedly found to have off target effects on the epigenome even if these do not lead to transcriptional changes that can be observed by RNA-sequencing. Furthermore, other papers using Tet1 dCas9 fusions have found that independent of the gRNA when expressed in cells these tools can lead to the loss of methylation from loci across the genome (including the locus they wish to target). Thus, I would strongly encourage the authors to perform the above WGBS experiment with all 3

conditions and report their findings. Given the previously published data (references provided in my previous reviews plus see article in Nature Communications 2021 from Sapozhnikov and Szyf; <https://doi.org/10.1038/s41467-021-25991-9>) I anticipate that the authors will find that their TETact system causes off target epigenome modifications as compared to the control cells without TETact, but it is critical that they report this in their paper as others who are not experts in the field and are naïve to the nuance of these tools will not be aware of this caveat and thus will not be aware of its limitations.

Reply:

We have now performed a global DNA methylation analysis (enzymatic methyl-seq) on WT A20, TETact with control guide and TETact with CD4-targeting sgRNA. Global analyses did not yield any statistically significant differentially methylated regions (DMRs) between any samples suggesting that TETact does not lead to widespread off-target demethylation (Supp Table 2). We then focussed on promoter regions and while the samples again had similar profiles (Fig. 2e, Supplementary Fig. 4c) we did reveal alterations between the A20 and the TETact containing cell lines (Supplementary Fig. 4c, Supp Table 3). However, when comparing the TETact control with CD4-targeted TETact we found only the CD4 promoter as the single DMR (Fig. 2e, Supp Table 3).

Reviewers' Comments:

Reviewer #2:

Remarks to the Author:

The reviewers have done a good job addressing my comments. I have no further comments.